# Generative Topographic Mapping of the Docking Conformational Space

**DOI:** 10.3390/molecules24122269

**Published:** 2019-06-18

**Authors:** Dragos Horvath, Gilles Marcou, Alexandre Varnek

**Affiliations:** Laboratoire de Chemoinformatique, UMR7140 CNRS/Univ. of Strasbourg, 1, rue Blaise Pascal, 67000 Strasbourg, France; g.marcou@unistra.fr (G.M.); varnek@unistra.fr (A.V.)

**Keywords:** docking, generative topographic mapping, conformational space maps, contact fingerprints

## Abstract

Following previous efforts to render the Conformational Space (CS) of flexible compounds by Generative Topographic Mapping (GTM), this polyvalent mapping technique is here adapted to the docking problem. Contact fingerprints (CF) characterize ligands from the perspective of the binding site by monitoring protein atoms that are “touched” by those of the ligand. A “Contact” (CF) map was built by GTM-driven dimensionality reduction of the CF vector space. Alternatively, a “Hybrid” (Hy) map used a composite descriptor of CFs concatenated with ligand fragment descriptors. These maps indirectly represent the active site and integrate the binding information of multiple ligands. The concept is illustrated by a docking study into the ATP-binding site of CDK2, using the S4MPLE program to generate thousands of poses for each ligand. Both maps were challenged to (1) Discriminate native from non-native ligand poses, e.g., create RMSD-landscapes “colored” by the conformer ensemble of ligands of known binding modes in order to highlight “native” map zones (poses with RMSD to PDB structures < 2Å). Then, projection of poses of other ligands on such landscapes might serve to predict those falling in native zones as being well-docked. (2) Distinguish ligands–characterized by their ensemble of conformers–by their potency, e.g., testing the hypotheses whether zones privileged by potent binders are clearly separated from the ones preferred by decoys on the maps. Hybrid maps were better in both challenges and outperformed the classical energy and individual contact satisfaction scores in discriminating ligands by potency. Moreover, the intuitive visualization and analysis of docking CS may, as already mentioned, have several applications–from highlighting of key contacts to monitoring docking calculation convergence.

## 1. Introduction

In two recent publications [1,2], the non-linear dimension reduction procedure called Generative Topographic Mapping (GTM) [3,4,5,6] has been adapted to visualize and analyze the abstract, (3*N*-6)-dimensional Conformational Space (CS) of arbitrary N-atomic molecules up to the size of polypeptides with more than 10 residues. It has been shown that generative topographic maps of CS are not only intuitive, supporting the visualization of key features and intramolecular interactions, but actually support construction of quantitative, predictive landscapes of conformational properties (energy, RMSD versus the native fold, etc.). This is a key advantage of the approach over “classical” state-of-the-art methods for CS analysis, as discussed previously [1,2].

GTM-based mapping of conformers is an excellent tool for comparing conformer sets (by monitoring the degree of overlap of the map zones they populate) without requiring any prohibitively expensive explicit pairwise conformer similarity scoring. It this can be used to control of convergence of stochastic conformational sampling simulations, which need to be run until they can be shown to reproduce the same CS coverage pattern on the map.

GTMs are a generalization of Kohonen maps [7]: in both, projection is done onto a latent 2D space defined as a (herein, squared) grid of nodes. In GTM, however, the assignment of an item is shared, with different weights (technically “responsibilities”) between several nodes. These are the key variables used to create property landscapes, by “coloring” each node according to the responsibility-weighted mean of the properties of resident items (here: conformers), and to predict the property of other conformers as like-wise weighted mean of values of the nodes it is residing.

This paper explores novel alternatives to analyze the CS of docking problems by means of GTM, all while creating a more general context enabling the direct comparison of (putative) ligands of a given active site. Pending further generalization to computational chemogenomic problems [8], the approach advocated here is to build dedicated maps for a given protein site, based on site-annotated contact fingerprint (CF) vectors, where the vector element CF_i_ monitors the number of ligand atoms (generic, or of specified type) within contact distance to (arbitrarily numbered) site atom i. CFs are a generalization of the popular concept of docking interaction fingerprints [9,10]–with the notable difference that all the solvent-exposed site atoms i are accounted for, and not only the “hot spots” known to host key interactions with the ligand. Contacts with ligand hydrogen bond donors, acceptors or high partial-charge ligand atoms are being monitored explicitly, by dedicated vector elements. Also, the CF vector explicitly monitors both direct contacts (corresponding to touching van der Waals spheres) and long-range contacts (corresponding to putative water-displacing interactions). As the Methods section will show, the final length of the CF vector represents the number of solvent-accessible active site wall atoms × types of ligand atoms in contact (any, hydrogen bond partner, charged) × two contact satisfaction distance ranges (direct, long-range).

Thus, a ligand pose will be essentially described only by means of the list of site atoms to which it makes (generic or specific types of) contacts.Intra-ligand contacts (or any other kind of ligand constitutional information) will not be captured. Therefore, CFs of different ligands are de facto describing a common, high-dimensional CF vector space. Specific ligand information can be easily added by concatenating any ligand-specific descriptor terms to the CF vector, in order to form a “Hybrid” (Hy) space. ISIDA fragment counts are information-rich descriptors that may be tuned to capture chemical information of various nature (including pH-sensitive pharmacophore typing), and the selection procedures of best-suited fragmentation schemes are routinely used to solve QSAR problems. Therefore, these fragment counts were used as source of ligand-specific information in Hy vectors. Hy vectors of ligand poses consist of the above-mentioned CF vector defining the pose concatenated to the (geometry-independent) ISIDA fragment count defining the ligand. Therefore, unlike in CF space, two distinct ligands establishing contacts of similar nature with the same site anchor atoms will nevertheless map to distinct areas of Hy space, depending on their intrinsic structural dissimilarity. The specific fragmentation scheme yielding best results in this context was fine-tuned as described in Methods.

Cyclin-dependent Kinase 2 (CDK2, e.g., P24941 by Uniprot Accession ID) has been selected for this study, given the richness of co-crystallized diverse binders of the ATP site in the Protein Data Bank [11], as well as the richness of inhibitors with measured thermodynamic instability constants, reported in the ChEMBL database. A subset of 84 ATP site binders from the PDB, showing no significant induced-fit modifications of the active site was selected. The active site geometry of 1CKP was used throughout this work. The cross-docking (respectively redocking, in the case of the 1CKP ligand) of all the 84 ligands into this 1CKP active site could be achieved with good results. Further ChEMBL [12] ligands, with unknown “native” binding modes (excluding the abovementioned 84) and DUD CDK2-decoys were also docked.

Docking was carried out according to a “reinforced” protocol using S4MPLE [13]. The approach is based on the previously described workflow [14]. Reinforcement consisted in repeating the last stage, the actual docking calculation consisting of 500 generations of evolutionary algorithm, until five such runs are found to converge within 1 kcal/mol with respect to the absolute best docked energy levels found so far. If this criterion is not achieved during 20 repeats, the procedure is terminated due to time constraints and analysis proceeds with so-far harvested poses. All unique poses found during these repeated runs and within a 50 kcal/mol energy window with respect to the lowest sampled energy level were kept. A subset of 200 most stable and diverse poses were selected from the above pool, for each ligand. For all poses of PDB ligands, RMSD with respect to the native pose and site-ligand interaction energies were calculated a posteriori with S4MPLE.

The manifold of the CF map was tuned by the already described evolutionary procedure for building “multitask” GTMs [15], using a random subset of 1CKP poses as Frame Set (FS—the pool of items used to fit the manifold in descriptor space). Since the choice of descriptor space is not a degree of freedom in this scenario using “naked” CF vectors, optimized parameters include only map-specific terms. Map quality was assessed in terms of the manifold ability to simultaneously accommodate two predictive landscapes describing the behavior of the 1CKP ligand: of RMSD to native state and of ligand-site interaction energy, respectively. Such landscapes are iteratively “colored” by projecting 2/3 of 1CKP poses, then used to predict RMSD and respectively energy of the left-out tier. Cross-validated determination coefficients Q^2^ were thus estimated for both properties and their mean served as fitness score for map selection.

Building the “Hybrid” map followed the same procedure, except that “frame” and “coloring” items stemmed from the combined pools of diverse poses of one (random) half of the 84 PDB ligands. Hy descriptors can be obtained by concatenating CF vectors with arbitrary ISIDA fragment counts–therefore, the descriptor selection mechanism supported by the evolutionary map optimizer served to pick the optimal ISIDA scheme in completion to CF descriptors.

The present study addresses three distinct applications of these docking space maps:(1)Assessing the convergence/reproducibility of the docking protocol. To this purpose, the docking of the 84 PDB ligands was repeated, and the pose families harvested by these in any way identical “B” runs were compared to the initial “A” output by assessing the degree of overlap of the covered map zones.(2)Native-state prediction challenge: Here, we assess whether projection of a ligand pose on a docking map landscape colored by RMSD information is able to predict native states. The native-likeness of new ligand poses could be predicted by positioning them onto the landscape created on hand of previously sampled, native- and non-native-like conformations. Less trivial, poses of a new ligand could be placed onto the landscapes drawn by other ligands of known native poses. Unless the new ligand adopts a significantly different binding mode, maps drawn by reference ligands might be relevant predictors of its native-like poses. So far, “new” ligands were the other co-crystallized species, allowing a posteriori validation of such predictions, in order to assess their actual prospective usefulness.(3)Ligand prioritization challenge: it consists in discriminating between binders and non-binders to a given protein site. Docking energy differences (or scoring functions) are far from perfect [14]. In the present work, a ligand can be represented by the pool of its poses, e.g., a cumulated responsibility pattern on the map. Thus, ligands (seen as conformer libraries) can be perfectly well compared to each other on GTMs–a widely used library comparison tool [16]. In as far as the shape of cumulated responsibility “clouds” of active ligands significantly differs from the ones of decoys, docking maps are able to host ligand activity or activity class landscapes. Projection of new ligands (after harvesting their representative pool of poses by docking) onto such landscapes might ideally lead to a quantitative prediction of affinity–or at least to a correct ranking of top actives over decoys, at better rates than provided by docking scores, or by classical exploitation of docking interaction fingerprints.

## 2. Results and Discussion

### 2.1. The CF Map

The best CF map found by the evolutionary parameter optimizer featured a 29 × 29 grid of nodes, with a manifold described by 144 radial basis functions of width parameter 1.8 and a regularization coefficient of 6.61. This map supports highly predictive property landscapes for both pose RMSD values and ligand-site interaction energy levels. In 3-fold cross-validation, prediction of the RMSD of iteratively left-out poses based on the landscape “colored” by the other two tiers of docked conformers of the “driving” ligand 1CKP returned Q^2^ values around 0.96, corresponding to a root-mean-squared error of 0.35 Å. In terms of interaction energies, Q^2^ values around 0.75 match a root-mean-squared error of about 2.6 kcal/mol. This proves that the map is able to provide a coherent description of the CS of 1CKP docking problem. In Figure 1 below, the left-most landscape (a:RMSD) is colored by the mean RMSD of resident conformers. Read areas are populated by native-like conformers (RMSD < 2 Å), blue means the residents are very far from the native pose (RMSD ≥ 6 Å) whilst the intermediate colors are inhabited by conformers of intermediate RMSD values, as annotated on the spectrum bar associated to the plots. It should be pointed out that “reading” an RMSD value from a map by color matching is not straightforward, even when aided by spectrum bars (transparency modulation by local density also alters the perception of colors). Landscape plots are excellent tools to illustrate generic trends in the data–but for quantitative interpretations these trends should be captured by statistical indices. Alternatively, interactive “reading” of local values by mouse-picking interesting zones on the plot can be achieved with dedicated software. For these reasons, spectrum bars will not be systematically provided for all landscape plots in this work.

Note that, in principle, on GTM property landscapes, intermediate colors may correspond to either zones populated by items of intermediate property values (desirable), or to problematic zones where items with both high and low property values cohabitate. The latter scenario can however be excluded here, given the very high cross-validation score. The other two landscapes are fuzzy classification landscapes, reporting how two distinct classes of poses are positioned with respect to each other on the map. In the middle landscape (b:Contact), the “blue” class represents all conformers in which the ligand is positioned such as to establish direct contacts (of any nature) with both the carbonyl and the amide hydrogen of key residue Leu 83. By contrast, areas in red are populated by conformers in which at least one of the two key atoms is far from any ligand atom. Intermediate colors are transition zones of CS where poses of both types coexist. Eventually, the right-most landscape (c:H-bond) distinguishes between poses establishing the characteristic bidentate hydrogen bond to Leu 83 (blue) versus conformers which fail to simultaneously satisfy both hydrogen bonds. The alignment of the landscapes illustrates some interesting insights. Trivially, all the red areas of (b) must remain red in the more specific hydrogen bonding landscape (c): each hydrogen bond is a contact, but not each contact is a hydrogen bond. Interestingly, however, areas with satisfied generic contacts that are not hydrogen bonds–blue areas of (b) “turning” red in (c)–exist but are very rare. Poses with 1CKP being deeply enough buried into the site in order to establish contacts with both Leu 83 key atoms are in their vast majority hydrogen bonding to the key residue (any other kind of van der Waals contact carrying an implicit desolvation penalty). Unsurprisingly, the CS zone corresponding to native-like poses (red on left-most plot) matches the area with very high degrees of satisfaction of the bidentate H bond–a hallmark of potent CDK2 inhibitors. Interestingly, non-native like poses with both hydrogen bonds being satisfied can be found in the North-West and North-East corners of the map.

The CF map thus represents the 2D projection of the CF vector subspace containing all the possible site-ligand interaction patterns observed upon docking of the 1CKP ligand. It is unclear whether this GTM model may also be a relevant support for describing CF interaction patterns of ligands other than 1CKP.

Projection of the CF patterns visited by the other ligands (in Figure 2, this is exemplified by RMSD landscapes) shows that the other ligands also ensure a rather exhaustive coverage of the map. In many cases, the map is perfectly able to localize the (red) native-like CS zone by contrast to other poses, so that many of the RMSD landscapes of the other ligands can be successfully cross-validated: for 20 ligands (including 1CKP) Q^2^ > 0.8, 21 ligands have 0.8 ≥ Q^2^ > 0.6, and 22 more still pass the cross-validation test at 0.6 ≥ Q^2^ > 0.4, for example 1H0V (0.46). RMSD landscapes of remaining 21 targets fail to cross-validate (including 1H0W and 2V0D)–which however does not automatically imply that the CF map is not relevant for those ligands. Indeed, the cross-validation propensity is also strongly affected by the relative sparseness of native-like poses for certain ligands: whilst 1CKP features 1032 native-like poses out of the sampled 6107 (17%), in 1H0W this ratio drops to 620/6476 = 9.5%, and to 94/7075 = 1.3% for 2V0D.

Visual inspection of ligand RMSD landscapes as exemplified in Figure 2 reveals a certain number of recurrent patterns in the plots. Examples in the Figure were actually selected to represent two recurrent visual patterns in the columns one and two, while examples of column three are distinct. At a heuristic level, RMSD landscapes perceived as similar seem indeed to correspond to similar structures: the central column, features three congeneric ligands based on a common scaffold. Structures in the left-hand column are not necessarily similar in the sense of high Tanimoto scores or common substructures, but they are similar in terms of their relatively low complexity within the CDK2 ligand family.

### 2.2. The Hybrid (Hy) Map

The optimal Hy map features a grid of 46 × 46 nodes spanned by 441 radial basis functions of width 0.3, and a regularization coefficient of 0.012. The map optimization procedure selected, out of the considered ISIDA fragmentation schemes, the IIA-FC-1-2 scheme to be the best suited to complement the CF vectors. This scheme corresponds to a descriptor counting the occurrence of circular fragments including first and second coordination spheres around the central atom, with atoms being labeled by chemical element, including flags for carriers of formal charge. Except for this latter detail, IIA-FC-1-2 are actually conceptually closely related to the popular Morgan fingerprints.

Trained on the basis of a collection of diverse poses from randomly selected ligands, this map is, unsurprisingly, of significantly higher resolution compared to the CF map. In terms of cross-validation propensities, the site-ligand interaction energy landscape associated to the Hy map obtains an excellent Q^2^ = 0.88, while the RMSD landscape reaches Q^2^ = 0.73. As the initial Hy descriptor space is a concatenation of both CFs and ligand-specific ISIDA descriptors, the conformer pool of any given ligand will be confined to a specific map zone, constrained by the ISIDA terms (Figure 3). Similar ligands will cover overlapping zones, while radically different ligands display non-overlapping density patterns.

### 2.3. Reproducibility of Docking Calculations

The three ligands with the worst Irreproducibility indices IRI (85%) (monitoring cumulated responsibilities on nodes populated to an extent of 85% or more by conformers visited by only one of the two docking simulations–see Methods section) are shown in Figure 4, next to the CF maps of the conformer populations returned by the two docking runs. By contrast, an example of near-perfect reproducibility in terms of CS coverage is also provided.

First, it is important to note that overall reproducibility is excellent: at worst, less than 5% of generated conformers fall into CS zones that were not reproducibly visited. Even though the area in dark red in the left-most landscape covers a significant part of the map, it mostly corresponds to low-density CS zones: even the single one docking run visiting it returns a negligible number of therein residing conformers. Note that in IRI (85%) the zones counting as not being properly revisited by the alternative run may still contain up to 15% of conformers from the latter. The main attraction basins in CS clearly are well explored by both runs, for all ligands. This is consistent with the key observation that native-like structures were always generated, for all of the 84 ligands–however, they were not systematically top-ranked in terms of energy (vide infra).

Two of the three “difficult” ligands are amongst the most complex molecules, with five or more rotatable bonds, and other similar structures display IRI (85%) values above 2%. By contrast, the rather simple 1VYZ and other similar ligands are consistently found at the “easy” extreme of the reproducibility scale. The case of 3TIY is however more interesting: the ligand is small and rigid, so internal degrees of freedom cannot explain its presence amongst the less reproducible cases. In its native pose, 3TIY does indeed seem to form the characteristic fork of hydrogen bonds with Leu 83, albeit using atypical anchoring groups (phenol -OH). In Figure 5, its binding mode is shown by contrast to 1VYZ, which also features the pair of key H bonds, yet this time involving the “classical” anchoring groups of kinase inhibitors–here, the bidentate aminopyrazole moiety.

The different behavior of the two ligands most likely stems from the fact that the small 3TIY will find a significant freedom to roam and explore multiple poses in the active site, which are not significantly higher in energy than the native pose (at least, they are not perceived as being irrelevant high-energy poses according to the AMBER/GAFF energy function of S4MPLE). While 1VYZ has only one possibility to satisfy the key H bonding pair, 3TIY allows for alternative poses involving other oxygen atoms on its scaffold. Thus, the multiple putative bound states of 3TIY are the source of its higher (thus intrinsically more difficult to reproducibly visit) CS volume. Binding of 3TIY should accordingly be enthalpically less favorized, but entropically less penalized than other CDK2 ligands. We did unfortunately not find any thermodynamic data to support or falsify this prediction.

Interestingly, the docking protocol requesting at least five independent evolutionary docking simulations to converge within 1 kcal/mol from the best-so-far docked energy minimum offers an alternative, energy-oriented point of view to reproducibility of docking calculations. By this criterion, the easiest-to-dock ligands will require only five repeats, returning every time the same bottom energy value within 1 kcal/mol. In the worst cases, the criterion will not be fulfilled even after the maximum allowed number of 20 trials. Thus, the number of simulations performed for each ligand is per se an indicator of the facility to reproduce the bottom energy level of a ligand. However, as Figure 6 shows, the facility to achieve convergence of the minimal energy and the facility to achieve reproducible coverage of the available CS volume are not correlated.

As the independent “docking runs” consist of the ensemble of independent simulations needed to pass the energy convergence criterion, it is expected to see longer docking runs, consisting of more of repeated simulations, implicitly become more reproducible in terms of CS coverage and score lower IRI values. Figure 6 appears indeed to outline this however, very weak trend – whilst the specifics of the energy landscape of each docking problem appear to ultimately control the ligand behavior with respect to the two convergence criteria. By highlighting in yellow all the ligands for which the most stable pose is native-like (RMSD < 2Å with respect to the PDB structure), it can be seen that difficulty to reproducibly descend to the bottom of the docking energy landscape (more than 10 simulations needed) is the hallmark of ligands in which this hard-to-reach absolute energy minimum eventually reveals itself as not native-like. There are two exceptions to this ad-hoc “rule”: already mentioned 1Y91 and analogous 3NS9 (based on the same isopropyl-substituted heterocyclic scaffold). These two compounds have rugged, difficult-to-sample but well-behaved energy landscapes (energy minimum is the native-like pose).

In the other concerned cases, the relative difficulty to sample the absolute energy minima does not prevent S4MPLE to easily and reproducibly revisit higher-energy CS zones, including those harboring the native-like poses. Thus, the very narrow, hard-to-find absolute minima of these energy landscape are either artefacts of the AMBER/GAFF parameterization of S4MPLE–or perhaps real minima that might correspond to the ligand binding mode at very low temperatures but are too narrow to be populated under physiological conditions. Again, we do not dispose of experimental evidence in favor or against such hypotheses. By contrast, the most straightforward explanation for ligands which easily and reproducibly converge towards non-native minima are force-field parameterization flaws. It is not the scope of this article to provide a detailed follow-up of all hypotheses that could be formulated here. We merely wished, so far, to exemplify how the monitoring of convergence behavior combining an energy and a CS coverage criterion may let the medicinal chemist highlight ligands with specific behaviors.

### 2.4. RMSD Landscapes and Predictability of Native-like Poses

As detailed in Section 3.6, the possibility to use GTM models to predict the RMSD of any given pose of some new ligand (i.e., not used to calibrate the predictor landscape) has been investigated for both CF and Hy maps. With the CF map, RMSD landscapes obtained by projection of the ensemble of docked poses of any co-crystallized “explorer” ligand may be used to predict the RMSD of some pose of any other ligand (co-crystallized or not). In this retrospective study, of course, the remaining 83 co-crystallized ligands are used for test, in order to be able to compare predicted and actual RMSD of the poses. This comparison returns a status label defining the ability of the RMSD landscape of explorer ligand e to recognize the native-like poses of the test ligand, possible verdicts being Failure (F), Moderate success (M), Ranking success (R) or Quantitative prediction success (Q). Please refer to the mentioned paragraph for details. Results, with each one of the 84 compounds iteratively playing the role of the “explorer” challenged to predict the remaining ones is shown in Figure 7.

For example, the CF-based RMSD landscape of ligand 2B53 is able to provide quantitative prediction of RMSD poses of nine other ligands. This landscape is the one with most successes of the Q category. The structure of the “explorer” 2B53 is shown (Figure 8) next to the nine ligands it is able to provide with quantitative RMSD predictions. Intriguingly, these are not obviously similar to the “explorer”. Note that 1CKP is not the explorer with most quantitative successes (it scores four, nonetheless)–but it is one of the ligands that can be quantitatively predicted by means of the 2B53 landscape.

Several other “explorers” of the CF space generate landscapes able to correctly rank the native poses of more than half of external ligands–even though the predicted RMSD values are not quantitatively correct, the poses at minimal predicted RMSD are indeed the native ones (R-type success). CDK2 ligands are chemically very diverse, to the point that there is no unique interaction pattern with the site to describe the binding of all, as could have been already intuited from Figure 2. The position of the native-like zone on the CF map is by no means unique, but rather ligand-dependent. There are ligands sharing a common interaction pattern (note that even the key bidentate H bond to Leu 83 is not systematically fulfilled by all the 84 co-crystallized ligands explored here). It is thus no surprise to fail predicting pose RMSDs for ligands with different site anchoring patterns that the “explorer” at the basis of the RMSD landscape.

Therefore, CF maps would–in the case of CDK2, at least–not be a viable option for predicting native poses out of a docked pool of conformations of a novel ligand, for the simple reason that prediction outcome would widely change depending on the used “explorer”–and there seems to be no simple rule allowing to predict which of the explorers would be most likely to be “compatible” in terms of RMSD landscapes.

With the Hy map, producing RMSD landscapes based on a single explorer clearly makes no sense, as external ligands are likely to project into “white spots” (i.e., out of applicability domain). A single RMSD landscape, based on the map selection set pool, was generated and used for prediction. Since half of the 84 ligands were represented by their diverse conformer subsets in the selection set pool, these selection ligands are allegedly easier to predict by the Hy map landscape (albeit the diverse subsets of 200 conformers/ligand injected at creation of the RMSD landscapes typically represent only some 3% or less of the many thousands of docked conformers subjected to prediction). Therefore, prediction success counts based on the Hy map were separately monitored for “selection” and “external” ligands, as shown in Figure 9.

With two failures and only one quantitative prediction success versus one and four, respectively, the prediction of “external” ligands is indeed–marginally–more challenging.

In docking, the baseline predictor of the native-likeness of poses is their energy level. If predicted poses are ranked in order of increasing S4MPLE energy, then native-likes should rank first, and a ROC curve of high AUC should be obtained. The herein developed methodology offers however the alternative to rank poses by predicted RMSD–as returned by projection on the Hy RMSD landscape. Is this ranking more or less likely to prioritize native-like poses, compared to the default S4MPLE energy based one? The answer, according to Figure 10 below, is positive. With three or four exceptions within the 42 external ligands (the selection subset is not shown), Hy-landscape-predicted RMSD is indeed a better criterion for native pose detection than energy. Sometimes, improvement exceeds 0.3 ROC AUC units–albeit S4MPLE energy is, per se, a quite robust ranking criterion, allowing better-than-random prioritization of native-like states for most of these ligands.

### 2.5. Virtual Screening Results

Prior to the analysis of GTM-based predicted ligand pK_i_ values as valid ranking criteria for prioritization of actives, Figure 11 shows that the classical virtual screening approaches used for benchmarking largely behave as expected. S4MPLE energy difference is clearly discriminant, in line with previous publications [14]–especially for the easier TvO task of discriminating between highly active compounds and the rest of weak binders and decoys. Also, in line with expectations–as such observations were the key incentive for the development of interaction fingerprint-based rescoring methodologies – the level of simultaneous satisfaction of contacts at both Leu 83 carbonyl and amide hydrogens is better discriminator than energy, in both scenarios. Note that ranking according to the intensity of a single interaction at a time (not shown) is also discriminant, but less strong than simultaneous interaction strength (here taken simply as the product of individual contact counts).

Map-inferred pK_pred_ values are clearly the best ranking criteria in this virtual screening benchmark, even though they are not quantitatively accurate predictions of actual pK_i_ values of test ligands (nonetheless, the determination coefficient of the two magnitudes reaches 0.3 for Hy map-based pK_pred_ at β = 0.3–marking the absolute best ROC AUC value in the TvO scenario). The better performance of the ligand structure-aware Hy map over the contact fingerprint-only CF approach is also expected. It might be argued that map-based approaches have an overhead advantage in terms of input information: they were trained (i.e., colored) with examples of contact patterns (and ligand substructure counts) associated to pK_i_ values, and they extracted “knowledge” about the patterns most likely met in actives versus the ones predominantly seen in inactives. This is exact–however, the focus on the rather successful criterion of the contacts in Leu 83 is also a form of knowledge extraction. Empirical, it was learned by humans observing kinase binding patterns. GTM-based models have now been shown to be able to complete this task even better, and without singling out any specific contact, no matter how important it is.

Last but not least, is the question can the results in Figure 11 provide any hint about the effective “temperature” value of a S4MPLE evolutionary docking simulation? The peak of the Hy map-based proficiency at β = 0.3 translates into increased ROC AUC in the TvO scenario and into an overall maximum for the pK_pred_ versus pK_i_ determination coefficient but does not trigger any improvement in the AvD scenario. Thus, it is likely due to a fortuitous improvement of the prediction of a few very active ligands, which happen to be more conveniently represented by the responsibilities of their few most stable poses. The general trend seen in the Figure suggests that (within the +50 kcal/mol window with respect to the lowest energy level – the selection criterion by which a pose is kept in the docked pool of geometries) plain averaging of responsibilities of all poses is the best possible strategy. Like in canonical NVT or NPT simulations, the relevance of a geometry is not to be defined by its potential energy, but by the number of times it has been sampled by a simulation. These findings match another, fully independent (but not very conclusive) attempt to assess the temperature of S4MPLE flexible docking simulations related to (unfortunately very few) enzymatic activity data [17].

## 3. Methods

The Methods section will first discuss compound curation, then briefly revisit already described technologies (docking with S4MPLE, GTM construction). Eventually, the new methodological developments (conception of CF/Hy vectors) will come in to focus. It terminates in describing how GTMs were used to address the challenges mentioned in the Introduction.

### 3.1. Compound Curation

All the PDB structures associated to the Uniprot code P24941 were downloaded from PDB and aligned in the PyMol interface (v. 1.9.0.0, Open-Source). Using the 1CKP ligand as center, a sphere of residues within 12 Å was defined, external atoms being deleted. PDB structures with no ligand in the selected ATP site were discarded. The active site to be used in docking was taken from the 1CKP protein structure (hydrogens were added and ionizable side chains set to charged status, using the MOE [18] tool). As required for S4MPLE docking, a series of five site “hot spots” was specified–atoms used as references for the randomized initial placement of the ligand in the site, but with no special status in docking energy evaluation. These included the carbonyl and >NH of Leu83 participating in the “iconic” pair of hydrogen bonds characteristic for kinase ligand binding, but also the Cβ of Phe80 and the carbonyls of Glu81 and Phe160. All protein site atoms were declared fixed in S4MPLE.

The 12 Å sphere of the “washed” protein was then again visualized in PyMol, against the overlaid poses of all the downloaded ligands. Ligands significantly clashing with the 1CKP protein geometry were discarded, since in the chosen rigid-site cross-docking strategy their native-like poses would not be accessible. 84 ligands were thus kept (see list in Appendix A), standardized according to the in-house procedure implemented on http://infochim.u-strasbg.fr/webserv/VSEngine.html and prepared for docking with S4MPLE as previously described [14]. This step includes automated hydrogen addition according to the preferred (ChemAxon plugin [19]-predicted) protonation states at pH = 7.4. Generation of an initial geometry by the ChemAxon conformer plugin [20] was specifically disabled, in order to preserve the initial alignment of the PDB ligand to the 1CKP site. This is further required to estimate the RMSD of docking poses, but does not impact the docking, which always departs from a population of randomized geometries, not including the input one. GAFF atom typing and, as needed, automated parameter generation complete the ligand preparation step.

In addition, ChEMBL ligands of CDK2 with reported thermodynamic instability constants K_i_ were downloaded. One hundred top binders (7.2 < pK_i_ < 9.0) and ninety one weak binders (4.7 < pK_i_ < 5.5) were selected for this study, in addition to one hundred one random CDK-2 decoys from the Directory of Useful Decoys (DUD) [21]. These 100 + 91 + 101 = 292 compounds with no experimentally known binding mode (PDB ligands reappearing amongst the downloaded ChEMBL series were explicitly discarded) were used in the Ligand prioritization challenge. They were also standardized and prepared for docking, as described above–including, however, the automated 3D structure generation step.

### 3.2. Docking with S4MPLE

Sampler For Multiple Protein and Ligand Entities (S4MPLE) [13,22,23] is a maximum-generality conformational sampling approach which handles docking as a peculiar conformational sampling problem of several independent species–here, the rigid protein and the flexible ligand. However, other scenarios are supported, from single molecule sampling (as herein used for exploring the states of the unbound ligand) to multi-species docking of several competing ligands, including displaceable water molecules into a receptor (which needs not to be a protein). The mol2 file used for the active site, as well as the hot_spots list for initial positioning of the ligands are given in the site directory of the Appendix A tar ball. Please refer to the already cited article [14] for a detailed outline of the S4MPLE rigid-site docking protocol, used in this work. This includes ligand standardization, initial geometry generation by ChemAxon and typing by the Generalized Amber Force Field (GAFF) [24] toolset, a first sampling of free ligand conformers in order to determine the minimal energy level of the unbound ligand, followed by a calibration run to fine tune conformational diversity thresholds applicable during the evolutionary algorithm-driven sampling of bound states, in preparation of this latter and final stage.

The herein introduced key change in the docking strategy concerns a reinforced convergence control: the key step of the procedure, i.e., the 1000-generation evolutionary docking simulation has herein been replaced by a series of shorter (500-generation) runs which, in exchange, will be repeated for 20 times, unless the Energy Convergence Criterion (ECC) is fulfilled. This ECC is assessed as follows: for all the so-far completed docking simulations, the lowest level of energy they have reached is extracted and simulations are ranked accordingly. If the simulations ranked 2–5 have attained an energy level not exceeding the level of the so-far best one by more than 1 kcal/mol, then ECC is fulfilled and further repeats of the docking simulation are cancelled.

Docked poses are selected by fusing outputs of all performed simulations and discarding duplicate geometries, as well as poses with an excess energy > 50 kcal/mol with respect to the above-mentioned lowest level. Also, a “diverse” representative subset of 200 poses is selected out of the harvested pool, by iteratively selecting the energetically next best conformation in the list, and then discarding all the following geometries that are similar to the kept one, in terms of S4MPLE conformational fingerprints.

Since S4MPLE does not include any a posteriori scoring function for docking poses, the S4MPLE docking score ΔE is taken as the energy difference between the most stable docked pose “ligand@site” and the most stable free ligand geometry, as obtained by S4MPLE simulation of the “naked” ligand:(1)$$\Delta E=min_{i}\langle E_{i}\rangle {}^{ligand@site}-min_{i}\langle E_{i}\rangle {}^{ligand}$$

### 3.3. Generative Topographic Mapping

Generative Topographic Mapping (GTM) is a non-linear mapping method [5,25], briefly revisited below, highlighting the recurring **keywords**. Herein, the in-house ISIDA GTM implementation was used.

In GTM, a point in the 2D **latent space** (e.g., on the map) corresponds to an image on the **manifold** embedded in the initial (CF of Hy) high-dimensional descriptor space. The manifold is defined by a mapping function **y (x; W**) assessed with the help of M radial basis functions (RBFs) of width w regularly distributed in LS. The latent space is covered by a squared **grid** of K **nodes** (K being a perfect square), each corresponding to a normal probability distribution centered on the manifold. This is used to compute the **responsibilities** R_kn_, here representing the degree of association between the CS point of conformer n and the node k. R_kn_ is the fuzzy-logics truth value of the statement “Docking pose n resides in node k”.

To obtain a GTM regression model, first a property landscape needs to be created. To this purpose, training (or “coloring”) items (docking poses with computed properties – here, RMSD to native pose, or site-ligand interaction energy) are located on the map. GTM nodes are assigned the responsibility-weighted mean values of input property values of their resident poses. Next, a subset of “test” poses is projected, and they acquire their predicted property values from the reference values of the neighboring nodes. The property landscape, graphically displaying the property averages per node, can be density-modulated by associating node color transparency to their cumulated responsibility level, with most populous nodes being intensely colored and near-empty ones fully transparent.

In GTM classification models, poses are labeled according to the class they are assigned to (for example 1 = non-native-like with RMSD ≥ 2Å, 2 = native-like; RMSD < 2Å). For each node k, the cumulated responsibility $CR_{k}\left(C\right)=\sum_{n\in C}R_{kn}\;$ of all residents belonging to a class C can be specifically computed. The class with maximal cumulated responsibility is said to “win” the node, which will thus be given the class-specific color (and will return class C as predicted value with probability R_kn_ for any external items residing on it). Alternative to above “winning class” landscapes, two-class problems can also be rendered as fuzzy classification landscapes, where the class labels “1” vs. “2” are treated as regular numeric properties and the calculated mean property per index provides a direct measure of the relative population ratio CR(1) vs. CR(2) in each node.

In this context, the optimal parameters for CF and Hy maps respectively were both selected using the already described evolutionary procedure [15]. For the CF map, the pool of docked poses of ligand 1CKP was used. There is no freedom in descriptor space selection–CF vectors of poses (vide infra) define the initial, high-dimensional space. The optimizer was allowed to select either of the three tiers of provided poses set as Frame Set (FS)–which serves to fit the flexible manifold into the initial space. GTM control parameters (node number K, number of RBF functions M, RBF width factor and regularization parameter) represented the other significant degrees of freedom. Two calculated properties were provided for each pose: RMSD to native geometry, and ligand-site interaction energy. However, poses with RMSD values above 6.0 Å were all assigned a maximum value of 6.0–in order to allow the model to focus on the relevant RMSD range (discriminating between “badly docked” at 6>RMSD > 2 and “quasi-undocked” poses at RMSD > 6 is both irrelevant and very easy to achieve, as the latter have near-empty contact fingerprints). For each property, associated landscapes on the manifold constructed according to current GTM parameters were iteratively built on randomized 2/3 of the pose pool and challenged to predict the left-out 1/3 of poses. This three-fold cross-validation was repeated three times, returning associated cross-validated determination coefficients Q^2^. Map fitness was estimated as the mean of these Q^2^ values, minus one standard deviation. Hy map fitting followed the same procedure, with two important modifications:(1)Frame and selection data were taken from the merged “diverse” conformer pools of 42 randomly selected ligands. The PDBligs.dat file in the Appendix A has these “training” ligands labeled “yes” in the third column of the file, by contrast to the 42 others left out for validation purposes (columns 1 and 2 report SMILES and source PDB entry, respectively).(2)Descriptor space selection is now enabled in the evolutionary map builder, as one hundred different Hy descriptor spaces were considered. These are concatenations of CF descriptors with either of one hundred types of different ISIDA fragment descriptors. These 100 ISIDA fragmentation schemes are the default options used as pool of meaningful descriptor choices for solving various chemoinformatics problems in our group.

### 3.4. S4MPLE Contact Fingerprints (CF)

By contrast to the conformational fingerprints used to monitor conformer population diversity in S4MPLE simulations, herein developed CF vectors are aimed at describing a ligand pose by means of the contacts it establishes with a given receptor site. Therefore, CF vectors are site-specific, vector elements being each associated to a binding site atom. Here, selected site atoms were defined by expanding the selection of overlaid ligands in PyMol by 4 Å, herewith including every site atom that was in contact distance to any of the considered ligands. Then, S4MPLE was used to calculate the solvent-accessible surface area of the (ligand-free) protein site. Site atoms included in above-mentioned selection but failing to expose more than 5% of their surface to the solvent were discarded. The remaining 156 “key” atoms (see keyContAtoms in the site directory of the archive in Appendix A) are basically the ones forming the active site “walls”. For each such key atom, two lists of ligand atoms are established, given a current pose. Ligand atoms enter the “direct” list if their distance to the key atom is shorter than the sum of van der Waals radii plus a tolerance interval of 0.5 Å. The “long-range” list includes all ligand atoms within previously mentioned cutoff plus a water molecule radius, i.e., extends the “direct” list by atoms close enough to putatively force the displacement of site-solvating waters. Contact lists are fuzzy-logics-based, i.e., a ligand atom is given a real-number association score to each list, varying from zero (distance >> cutoff) to 1 (distance << cutoff) according to a sigmoid with inflexion point at cutoff. From each list, five distinct descriptor terms are derived:Total number of contacts (sum of all association scores of ligand atoms to this list)Number of hydrogen bond donors amongst contacts (weighted by association scores)Number of hydrogen bond acceptors (weighted by association scores)Total partial charge in contact atoms (weighted by association scores)Sum of absolute values of partial charges, likewise weighted

Thus, with two contact lists, each generating five descriptor terms for each of the 156 key atoms, the nominal CF space dimensionality equals 1560 for the herein studied CDK2 docking problem.

### 3.5. Monitoring Convergence and Reproducibility of Docking Calculations

Docking space maps offer an alternative point of view to check reproducibility/convergence of docking calculations, independent of the ability to rediscover the so-far deepest energy well (as defined by the ECC in Section 3.2). Like already exemplified for the dipeptide sampling problem, it is possible to check whether poses obtained from a repeated run of the docking procedure will project into the same nodes acquiring the poses from the initial run. If so, rendering of a fuzzy class landscape in a blue/red polarized spectrum, with extreme colors representing poses of either runs will result in a green/yellow-dominated image, as the two sets homogeneously mix throughout the entire conformational space. By contrast, should some zone be visited only by one of the two runs, it will stand out in corresponding extreme colors red/blue. Note that visual inspection of such landscapes intuitively provides an estimate of reproducibility degree, expressed by the level of overlap of resulting pose pools. Nonetheless, it is important to keep in mind that map areas are not equally densely populated (population density being used to modulate transparency of the landscape colors) – extreme colors on display are proof of irreproducibility only if they concern highly populated areas. CS space zones of intrinsic low probability visited “by accident” in one simulation out of two do not indicate reproducibility problems. For this reason, intuitive plots will be in addition characterized by a quantitative Irreproducibility Index IRI(p%), representing the density accumulated in nodes with populations prioritarily (>p%) stemming from one run, reported to the integral density on the map (i.e., total number of poses in both runs). The CF map was employed in the Reproducibility study.

### 3.6. RMSD Landscapes and Predictability of Native-like Poses

Ligand RMSD of poses versus the PDB structure is straightforward to calculate, as the overlay is implicitly ensured by the common reference system of the rigid protein site used for docking. Therefore, given a pool of poses of a ligand, RMSD landscapes can be colored and reciprocally used to predict the RMSD of any new pose projected on the landscape. The “classical” RMSD threshold of 2.0 Å was chosen to distinguish native-like from non-native-like poses. Unless otherwise mentioned, all further reference to RMSD will tacitly imply the upper-bounded values at 6.0 Å, as discussed in Section 3.3.

The pose to predict may belong to the ligand which served to create the CF-map landscape. For this scenario, the RMSD-prediction propensity of maps was assessed by 3-fold cross-validation (landscape construction based on 2/3 of the docked pose pool, with prediction of remaining tier and thereupon based calculation of a cross-validated determination coefficient Q^2^). However, as maps were designed to accommodate any ligand, the RMSD landscape calibrated on hand of a co-crystallized ligand might be useful to predict native-likeness. Thus, it is possible to project poses of some test ligand l onto the landscape calibrated by an “explorer” ligand e. If the explorer has visited CS regions that are also relevant for l and the two small molecules share a common binding mode, then the landscape described by the explorer might serve to correctly predict the native-like states of the test compound. To this purpose, the complete matrix of cross-predictions was generated, using each of the 84 PDB ligands as “explorer”, then projecting the remaining 83 onto the generated RMSD landscape. Predicted RMSD were then compared to observed RMSD values, according to three different points of view: (1) quantitative correlation as expressed by the determination coefficient R^2^, (2) Balanced accuracy BA, expressing how well native-like (RMSD < 2) states were predicted as such, (3) ROC AUC value in prioritizing observed native-like states by ranking according to predicted RMSD. Given an explorer ligand, test ligands were labeled:(a)“Q”, “Quantitative” if all the three quality criteria exceed 0.75.(b)“R”, “Ranked well” if condition (a) is not fulfilled, but the ROC AUC value exceeds 0.8(c)“F”, “Failed” if none of the three criteria manages to exceed 0.6(d)“M”, “Moderate” for all the other situations with weak, yet better-than-random results.

For each explorer ligand, the number of test ligands achieving Q, R, M or F status is monitored. In addition, the ROC AUC values from ranking by predicted RMSD will be compared to the ones from ranking by energy, representing the default hypothesis in docking calculations: top stability poses are expected to be the native ones.

The Hy map is by definition created and “colored” on the basis of pooled poses from several ligands. The RMSD landscape created by the map selection set (vide supra) was likewise challenged to predict RMSD of the entire set of docking poses of the 84 ligands. Then, success rates were specifically counted for the ligands which participated in map selection, and completely external compounds, respectively.

### 3.7. Selective Recognition of Actives (Virtual Screening)

As the ultimate goal of docking calculations is the prioritization of active compounds out of a database of structures, and the use of contact fingerprints is known to significantly enhance success rates compared to the default ranking by docking scores, it is straightforward to ask whether CF-based maps may also be effective ligand/decoy discriminators. This problem was investigated on hand of the above-mentioned collection of 292 compounds including (roughly) 1/3 of potent binders, 1/3 of weak binders and 1/3 of DUD decoys. The challenge is to obtain Receiver Operating Characteristic (ROC) curves of maximal Area Under Curve (AUC) when ranking compounds by various calculated scores and expecting actives to be top-ranked. Practically, two scenarios were considered:discriminating the 100 potent actives on one hand, with respect to weakly actives an decoys on the other, i.e., the “Top versus Others” or TvO scenario,discriminating tested actives (strong and weak) from the 100 DUD decoys, i.e., the “Active versus Decoy” or AvD scenario.

For both scenarios, ROC curves were generated according to the following ranking criteria:ΔE: S4MPLE docking score, as given in Equation (1)LeuO: Average of the total number of direct contacts involving the Leu 83 carbonyl oxygen, with any ligand atomLeuNH: Idem, for the Leu 83 amide hydrogen.LeuTot: Average of the LeuO × LeuNH contact counts, marking how often ligand atoms are seen in the neighborhood of both site key spots.pK_pred_(β): GTM-based predicted pK_i_ value, based on quantitative activity landscapes supported by CF and respectively Hy maps, as detailed below, where β = (k_B_T)^−1^, T represents the (unphysical) temperature parameter of the evolutionary sampling process that lead to the docked conformer set and k_B_ is the Boltzmann constant.

Unlike in 2D-descriptor-based chemical spaces (where one compound is one projected item), the originality of the present approach consists in describing a compound by its ensemble of docked conformers. The GTM formalism is however perfectly adapted to render and compare cumulated responsibility patterns–describing for instance chemical libraries by the cumulated responsibilities of their chemical compound members. Here, a compound will be likewise rendered as a “library” of its docked conformers. However, unlike in library comparison, the average responsibility vector will be used here. Above, responsibilities of each conformer n contribute proportionally to the Boltzmann factor of the conformer, and the normalization with respect to the sum of these factors ensures that $\underset{k}{\sum }AR_{k}\left(C\right)=1$, i.e., the AR vector sums up to one as expected from the responsibility vector of a single item, and does not depend on the actual number n of conformers harvested for compound C. By modulating β, more stable conformations are allowed to have more impact on AR values: at high β (low T) AR will become equal to the responsibility vector of the single most stable conformer, whilst β = 0 means plain averaging of conformer contributions:(2)$$AR_{k}\left(C\right)=\sum_{n\in C}R_{kn}\exp\left(-\beta E_{n}\right)R_{kn}/\sum_{n\in C}\exp\left(-\beta E_{n}\right)$$

A ligand will thus be described by its AR vector, obtained by projecting its docked conformers on the map and taking the weighted average at user-chosen β value. To this purpose, a (random) half of the 292 compounds was used to “color” pK_i_ landscapes associating their AR vectors to their ChEMBL-reported pK_i_ values (for DUD decoys, pK_i_ = 3 was assumed). Then, the remaining 146 “test” compounds were at their turn projected onto these (β-and scenario-dependent) landscapes. Their predicted values pK_pred_(β) were determined and used as ROC curve generating (ranking) criterion.

When generating ROC curves, in the TvO scenario the potent binders were labelled “active” while weak binders and decoys formed the “inactive” class, whilst the AvD scenario had both potent and weak tested binders as “actives”, with DUD compounds being inactives. ROC AUC values will thus be reported for test compounds only–including ROC AUC scores with respect to the other “coloring set”-independent criteria ΔE, LeuO, LeuNH and LeuTot. Both color and test compounds are provided in the Appendix A which reports SMILES, pK_i_ value, activity class (“top”, “weak” or “decoy”) and eventually its status with respect to map-based virtual screening (“color” or “test”, respectively).

## 4. Conclusions

This work describes a procedure for non-linear (GTM-driven) mapping of the conformational space of a docking problem, as seen through the prism of site-ligand Contact Fingerprints (CF). To this purpose, the in-house S4MPLE tool was updated to include a contact fingerprint calculator, converting each saved docking pose into a CF vector with elements encoding intensities of the contacts of different kinds (generic, H bonding, partial charge-modulated, etc) at each of specified key atoms of the active site. A ligand can be thus characterized by the volume of CF space occupied by its selected poses. Such a description is however void of any explicit structural information about the atom types and connectivity of the ligand, CF descriptors were tentatively concatenated with ligand 2D ISIDA fragment counts, to generate an even higher-dimensional Hy space. An explicit search for the best candidate ISIDA fragmentation scheme was part of the Hy map optimization procedure, and the selected option was a circular fragment-based descriptors equivalent to Morgan fingerprints.

Neither the CF nor the Hy space may be directly analyzed, but require aggressive dimensionality reduction by GTM, in order to be converted into interpretable 2D maps. GTM parameters (and meta-parameters, like the above-mentioned ISIDA descriptor type for the Hy map) were optimized by an evolutionary procedure, where the objective consisted in maximizing the cross-validated prediction propensity of map-based RMSD and site-ligand interaction energy landscapes. The CF map was trained on the basis of the pool of conformers of a single CDK2 ligand (1CKP), while the Hy map based on a composite pool of diverse poses from 42 co-crystalized ligands. Both CF and Hy maps reached very high cross-validation scores and were thus used to explore various aspects of the docking process into the CDK2 site.

The CF map, in spite of being trained on contact fingerprints of a single ligand (1CKP), is able to accommodate projections of the other ligands, allowing in most of the cases for a clear outline of the “native-like” CS zone (including poses of RMSD < 2Å) by contrast to non-native geometries. It highlights the high binding mode diversity amongst the considered 84 co-crystallized CDK2 ligands, with several, non-overlapping native-like zones marking distinct binding modes. Ligands which so share a same native-like zone on the CF map may be structurally similar–albeit sometimes this degree of similarity is very low (small, “fragment-like” but otherwise chemically not obviously related ligands may, for example, have significantly overlapping “native-like” zones). It may thus be possible to predict native-likeness of poses of a new ligand by projecting its set of conformers onto the RMSD landscape of an “explorer” (a compound of known binding mode, used to probe the CF space)–but only if the two actually happen to share a common binding mode. Since this cannot be taken as granted, CF maps are not well suited for prospective prediction of native-like poses of new ligands.

By contrast, in Hy maps including explicit ligand structure information, chemically distinct ligands will span non-overlapping zones on the Hy map. Thus, projection of the conformer pool of a new ligand on a Hy map landscape, colored with information from a diverse set of reference compounds, will implicitly position it into the neighborhood populated by the chemically most relevant references, i.e., the ones most likely to share a same binding mode. The Hy map therefore provided very good results when challenged to prioritarily recognize native-like poses of the 42 external ligands not used to color the underlying RMSD landscape. The ability to rank native-like poses was remarkably better than the one of the default criterion expected to correlate with native-likeness: the docking energy level.

CF maps are, however, nicely suited to evaluate the convergence/reproducibility of docking calculations. Comparing the propensity of a repeated docking run to revisit the same map zones with the propensity of individual S4MPLE evolutionary simulations to reproducibly converge towards a same minimum energy, ligands with different behaviors can be singled out. While it is tempting to relate observed convergency behavior to the degree of ruggedness of the docking energy landscape and/or total volume of well-docked space (i.e., binding entropy), it is beyond the purpose of this work to experimentally check whether such relationships really exist.

Eventually, describing every ligand by its (putatively Boltzmann-weighted, at user-defined empirical β factor) mean responsibility over its docked states opens the way to use both CF and Hy maps as tools to compare different ligands, opening the door to Virtual Screening applications. To this purpose, 292 highly active, weakly active and DUD decoy compounds (the former two classes taken from ChEMBL, amongst CDK2 ligands with reported pK_i_ values, while the latter were assigned a default pK_i_ = 3) were evenly split into a “color” and a “test” set, respectively. The average responsibility vectors of color compounds, associated to their pK_i_ values, were used to generate predictive pK_i_ landscapes on both CF and Hy maps. Test compounds were projected thereon, and the herewith obtained pK_pred_ values served to rank prioritize the actives among them. By contrast to both ranking by S4MPLE docking energy difference and by degree of satisfaction of the key bidentate interaction with Leu 83, pK_pred_ values are clearly the most potent prioritizes of actives. This is irrespective on the specific scenario, considering only the top binders (pK_i_ > 7), or allowing all but DUD decoys to count as “actives” in generated ROC curves. This is not surprising, as the GTM-supported activity landscape embed a significant amount of structure-activity information extracted from the color set. However, the use of interaction fingerprints–like the focus on the Leu 83 interactions—is likewise exploiting prior knowledge, extracted by scientist who analyzed kinase-ligand X-ray structures. GTM-based analysis thus stands out as an excellent tool for–unbiased, e.g., not a priori focusing on specific site hot spots—extraction of preferred binding patterns.

Virtual Screening results were also an opportunity to address the question whether it makes sense to consider Boltzmann averaging of conformations produced by non-physical, evolutionary simulations like the one in S4MPLE. Unlike in molecular dynamics, there is no hint to suggest whether conformational sampling by mimicking a Darwinian evolutionary process will return a “microcanonical” or a “canonical” ensemble of geometries. While none of the two alternatives is probably true, the herein obtained results over the range of considered β values tends to show that Boltzmann weighing of conformers at room temperature is clearly detrimental, while plain averaging returns excellent results.

## Figures and Tables

**Figure 1 molecules-24-02269-f001:**
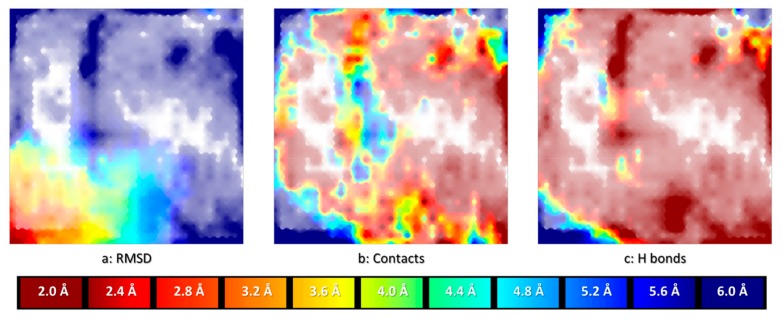
Aligned landscapes of the CF map, on which the docked ensemble of conformers of 1CKP was projected. The local density of residing conformers modulates color transparency and is the same in all plots. **a**: RMSD landscape–quantitative landscape with color encoding the RMSD of the poses with respect to the native structure (red = native-like zone, RMSD < 2Å, blue = non-native poses, RMSD ≥ 6Å, see spectrum bar below). **b**: fuzzy classification landscape overlapping the subpopulations of poses in which both key atoms (carbonyl O and amide H) of Leu 83 are simultaneously within direct contact distance to at least one of the ligand atoms (blue) versus poses in which at least one of the two key atoms is far from any ligand moiety (dark red), intermediate colors signaling that these subpopulations coexist in these “transition” areas. **c**: fuzzy classification landscape similar to (**b**), but specifically checking the satisfaction of a bidentate hydrogen bonding pattern to Leu 83.

**Figure 2 molecules-24-02269-f002:**
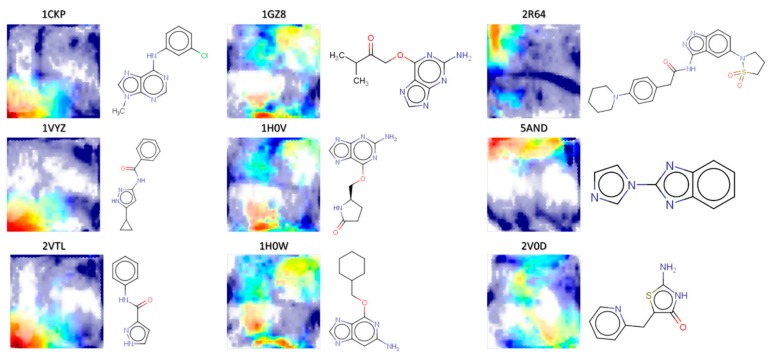
RMSD landscapes (red: native-like zone) of the docked conformer sets of various ligands on the CF map (including 1CKP, the ligand for which the map was built).

**Figure 3 molecules-24-02269-f003:**
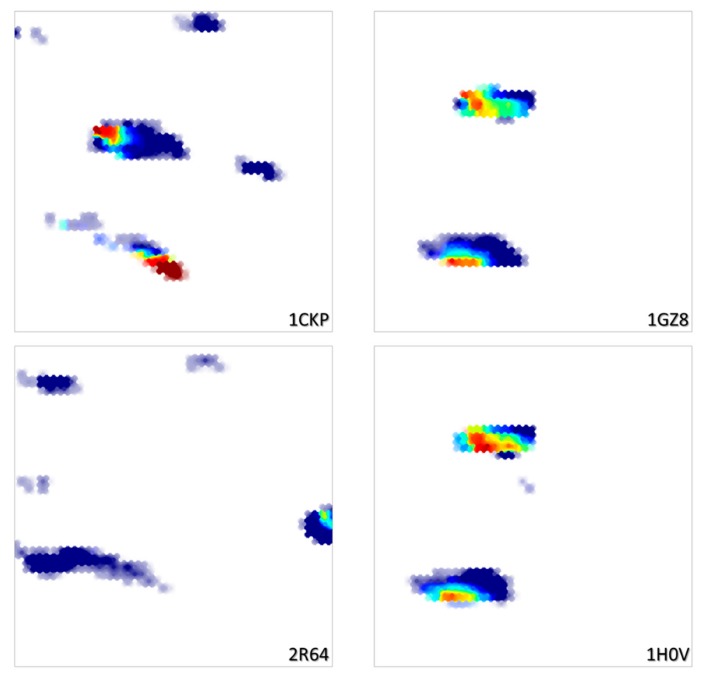
RMSD landscapes on the Hy map for four ligands also shown in Figure 2. Note how the similar ligands 1GZ8 and 1H0V cover the same map zones, whilst the structurally distinct 1CKP and 2R64 each claim their specific subdomains.

**Figure 4 molecules-24-02269-f004:**
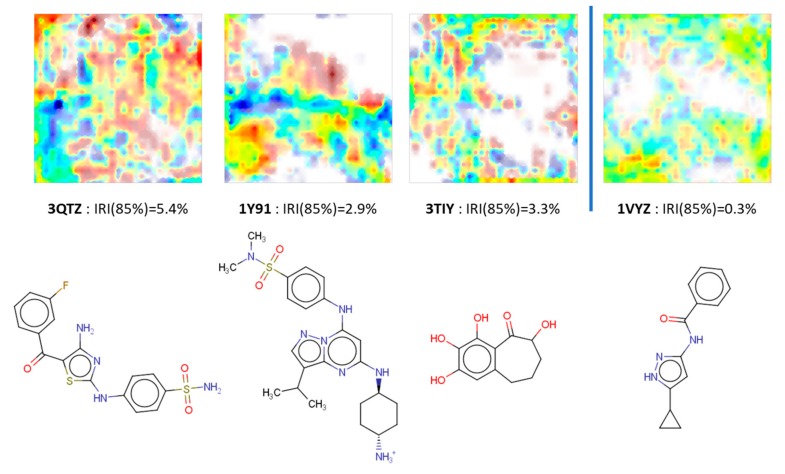
Ligands most and respectively (right-hand) least prone to fail visiting the entire conformational space during one docking run (as described in Methods). Fuzzy class landscapes based on the CF map highlight in extreme colors dark red or dark blue the CS zones visited by only one of the two repeated docking runs, whilst intermediate colors denote zones into which both runs found conformers. Transparency of the color is gradually increasing in less populated zones, producing the white spots in the “empty” zones, corresponding to site-ligand interaction patterns never observed for the given ligand. See Methods for the definition of the Irreproducibility Index IRI.

**Figure 5 molecules-24-02269-f005:**
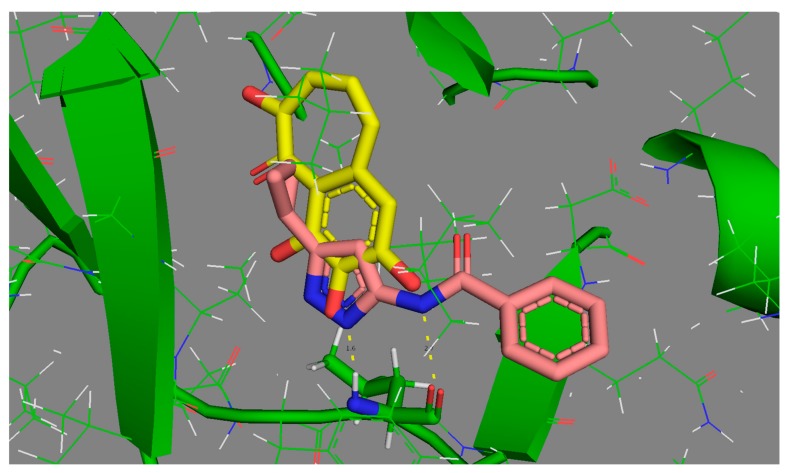
Comparative binding modes of easily reproducible 1VYZ (pink) versus “difficult” 3TIY (yellow carbon skeleton). Both feature the key hydrogen bonding “fork” to Leu 83 (green carbon skeleton in stick mode, at the bottom of the active site).

**Figure 6 molecules-24-02269-f006:**
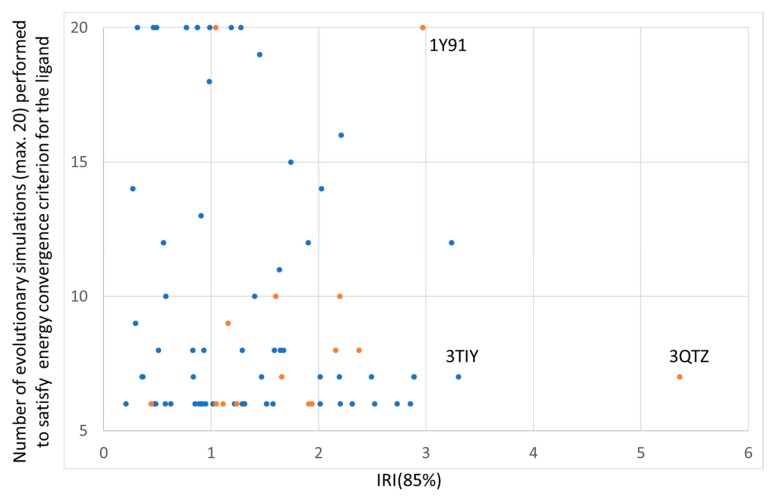
Simulation repeats required to reproducibly achieve best energy level, versus IRI index. On Y, the number of evolutionary simulations (limited to 20) performed by the S4MPLE docking script for each ligand in its attempt to reproduce at least five times a docking energy level within 1 kcal/mol from the known absolute optimum represents the facility to achieve convergence in terms of energy values. On X, the Irreproducibility index as estimated from the CF map measures the facility to reproduce CS coverage by distinct docking simulations. Each ligand is represented by a dot: in yellow, ligands for which the absolutely best pose in terms of energy is “native-like” (RMSD < 2 Å with respect to the PDB structure). In blue, the other ligands–for which native-like poses were sampled, but not top-ranked by S4MPLE docking energy.

**Figure 7 molecules-24-02269-f007:**
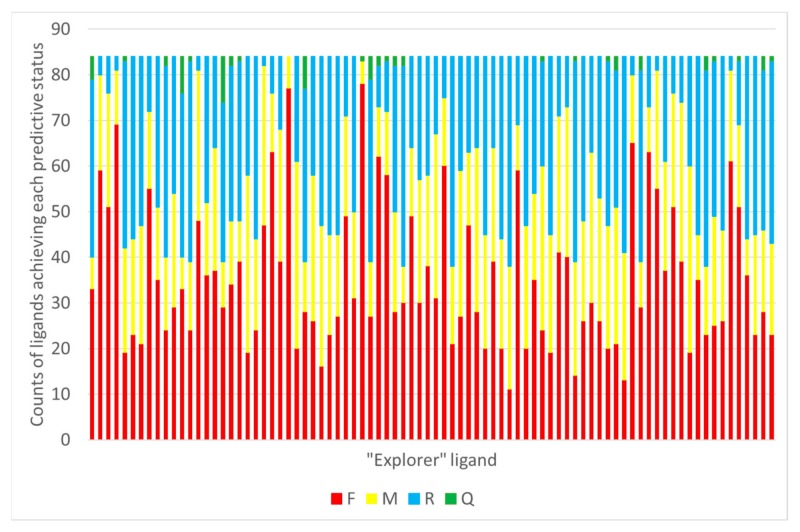
For each explorer ligand used to generate a CF-based RMSD landscape, this is challenged to predict RMSD of poses of remaining ligands. The status of each such prediction is assessed, and the total counts of Failed (F), Moderately successful (M), Ranking success (R) and Quantitative prediction RMSD success (Q) are reported on Y.

**Figure 8 molecules-24-02269-f008:**
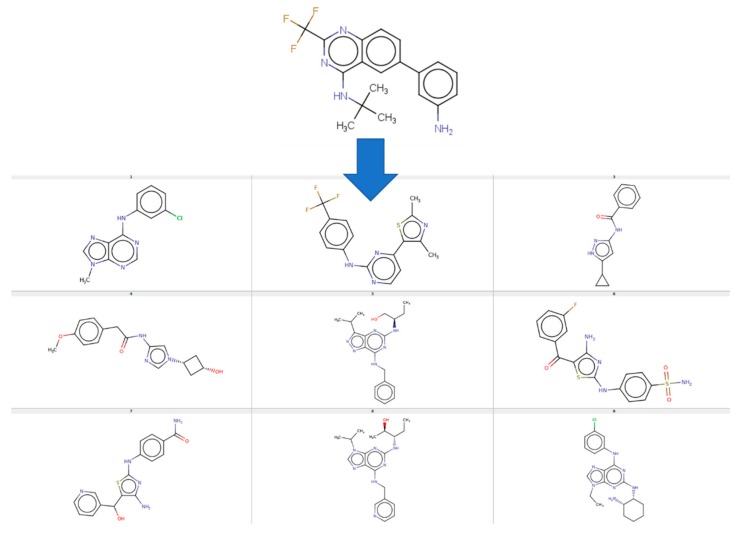
Structure of ligand 2B53, generating a RMSD landscape able to provide quantitative predictions for nine other–structurally not obviously related–ligands, shown below.

**Figure 9 molecules-24-02269-f009:**
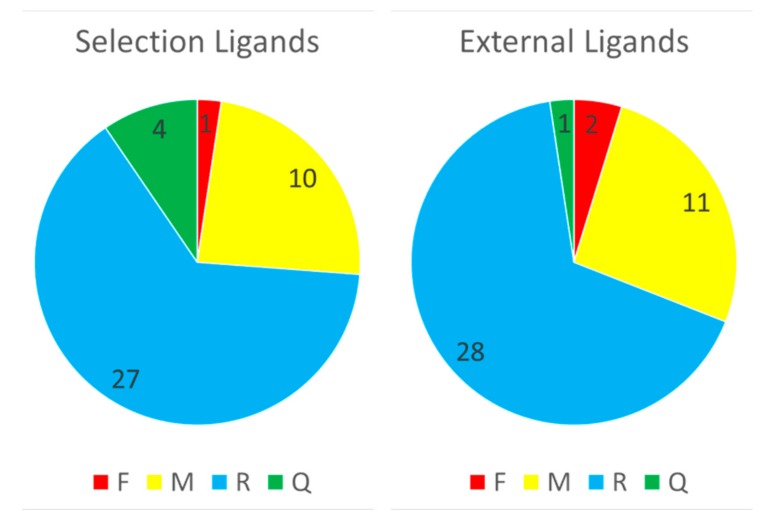
RMSD prediction success counts-by preestablished categories F, M, R, Q–for selection and external ligands, using the RMSD landscape on the Hy map. Numeric labels are numbers of ligands concerned.

**Figure 10 molecules-24-02269-f010:**
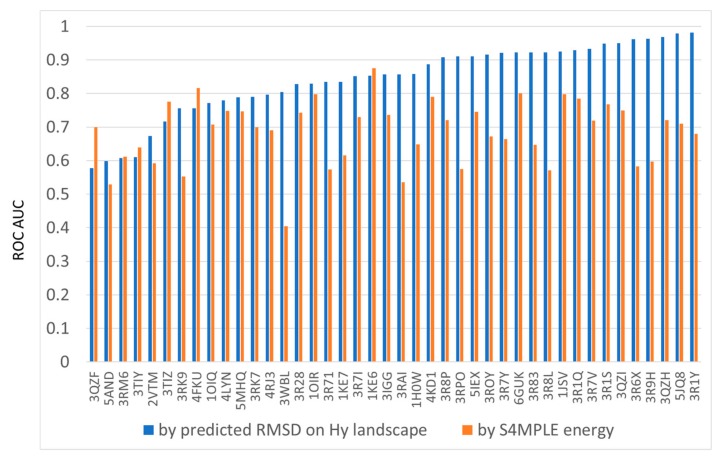
Comparative ROC AUC values obtained upon prioritizing the native-like poses of external ligands according to (in blue) RMSD values predicted by the Hy-based RMSD landscape versus (in orange) S4MPLE energy levels.

**Figure 11 molecules-24-02269-f011:**
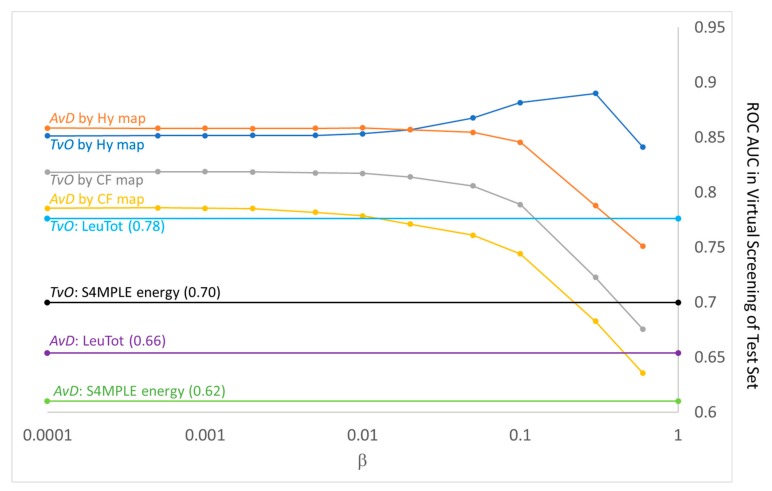
Comparison of ROC AUC levels in challenges to prioritize the “actives” in virtual screening scenarios TvO (Top-versus-Others) and AvD (Actives versus Decoys), as introduced in Section 3.7. All data refer to the 146 test compounds not serving to build the β-dependent activity landscapes on the CF and HY maps. The thereby generated β-dependent pK_pred_ values served to build β-dependent ROC curves labeled “scenario by map” above. By contrast, standard ligand ranking criteria such as S4MPLE energy and degree of satisfaction of simultaneous contacts at both key atoms =O and >NH of Leu 83 produce one single ROC curve per scenario, the AUC of which is rendered by corresponding horizontal lines.

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
