# Peer review of "Generative Topographic Mapping of the Docking Conformational Space"

_molecules, 2019, doi:10.3390/molecules24122269_

Round 1

Reviewer 1 Report

Horvath and co-workers describe the application of Contact Fingerprints (CF) built by their previous Generative Topographic Mapping (GTM) approach to docking calculations, which results in an improved outcome of the docking solutions. The use of CF has been successfully used in the past to improve docking results and, here, the authors have been further implemented it in their S4MPLE tool. The methodology shown in this manuscript is robust and it seems to have high reproducibility, which could highly benefit the medicinal chemistry field for the design and development of new ligands towards a defined target.

The manuscript is very well organized and it has been carefully written in a very comprehensive manner. Although the supplementary material was not accessible to this reviewer (it is only available upon request of the reader?), I find enough merit to publish this work in Molecules journal and I will highly recommend it for publication after minor revision.

Formatting of references should be revised. In addition, the date of accession is missing in references 11 and 19. Doi is also missing in several references.

Contour levels scale for the different landscapes on Figures 1-4 are missing.

Line 288: A short title for the legend in Figure 6 is missing.

Line 330: It seems that Figure 7 is missing at the end of the sentence “the remaining ones is shown in”

Line 431, a reference just accepted has not been included in the Reference section.

In Methods, line 598: the abbreviation LS has not been previously introduced in the manuscript.

Author Response

We wish to thank the reviewer for carefully proofreading the text. We did our best to respond to the issues and provide the suggested fixes whenever possible. Please find our detailed responses below:

Although the supplementary material was not accessible to this reviewer (it is only available upon request of the reader?), I find enough merit to publish this work in Molecules journal and I will highly recommend it for publication after minor revision.

Ø  This is strange – to our best recall, the SupMat.zip file had been uploaded at submission. There should have been no problem – it should be accessible to both reviewers and readers.

Formatting of references should be revised. In addition, the date of accession is missing in references 11 and 19. Doi is also missing in several references.

Ø  Unfortunately, this is true – albeit we used EndNote Web with the MDPI “official” style sheet, so… in principle this should have been as “official” a bibliographic list as one may hope for. No way – at a closer look we even found duplicate DOIs (the missing one tend to match older documents, for which there may not be any).

Contour levels scale for the different landscapes on Figures 1-4 are missing.

Ø  Yes, on purpose: in our experience, we found any attempts to “read” a quantitative value from the map by following the color scale to be often misleading (transparency modulation for density may interfere with the color perception). Therefore, in publications we only use maps to illustrate the main trends – we specify what the extreme colors mean, but we’d discourage the reader to attempt a detailed “decoding” of intermediate nuances. Instead, we provide quantitative criteria as rigorous backup, and use the plots to convey an intuitive picture for the numbers (IRI indices, in particular). This is now explicitly stated in the text – with a spectrum bar attached to Figure 1, to make the point. In practice, we do actually “read” the map, but use graphical interfaces to click on the spot of interest and return local characteristics.

Line 288: A short title for the legend in Figure 6 is missing.

Ø  Very good point. “Simulation repeats required to reproducibly achieve best energy level, versus IRI index” could be the short description; added.

Line 330: It seems that Figure 7 is missing at the end of the sentence “the remaining ones is shown in”

Ø  The reference in Word was missing. Thank you for having noticed that!

Line 431, a reference just accepted has not been included in the Reference section.

Ø  Indeed, thank you!

In Methods, line 598: the abbreviation LS has not been previously introduced in the manuscript.

Ø  Sorry, but in my word document there is no such abbreviation (LS) anywhere, though there is a CS (conformational space). Word2pdf artefact?

Reviewer 2 Report

This manuscript is very hard to read. I have to tell, that I am not an expert (and not very much interested in) Kohonen maps and related stuff, so I tried to understand this manuscript from a molecular modeling point of view, and was interested if the new method can provide solutions for the challenges of docking.

The abstract starts with a rather confusing and not clearly defined collection of non-standard terms of CF and Hy. There is an abbreviation (ISIDA) without any reference to its long name.

The introduction is extremely long with too many methodological details in the second half. After reading the introduction it is still not clear why GTM and CF were investigated for the docking problem.The goals are not clearly defined: there is a rather long and highly qualified text on "Assessing the convergence" and "a posteriori validation" but it does not tell too much for me. Probably, it is my fault.

Concerning the validation. I cannot appreciate that the 84 ATP site-binders show no significant induced fit modifications on the binding site (line 88, p. 2). Excluding ligands with induced fit obviously results in a biased training/validation. The language of the MS is also very confusing at such important points (line 90):  "cross‐docking (redocking,for the 1CKP ligand)" was "of" meant instead of "for"? At the top of page 3, there is a nice "Methods" section still inside the Introduction.

In the R & D, figs of various landscapes are extensively used. For me, it was impossible to interpret these landscapes and get the accurate (or at least some) physical meaning of the maps or the results in general. I know, the maps do not have such, but still... Again, there are confusing and/or missing definitions, also in the statistics etc.

"It is thus unclear, whether this GTM model may also be a relevant support for describing CF interaction patterns of ligands other than 1CKP.." (p. 5, line 191) Maybe, this is the only sentence in this MS, I can completely agree.

The web link on p. 14 does not work.

All-in-all, I would not say yes or no and do not feel authorized to rank this manuscript. However, it is not really worth reading, in its present form, I am afraid. Maybe, Kohonen map fans could enjoy it. (Please, do not send it to me for re-review.)

Author Response

The abstract starts with a rather confusing and not clearly defined collection of non-standard terms of CF and Hy. There is an abbreviation (ISIDA) without any reference to its long name.

Ø  Well, to our understanding an abstract is only meant to give an idea of what has been done – that is mapping of the conformational space using contact fingerprints CF (where the latter concept is not really new for the docking community, is it?) and their hybrid contact/ligand descriptors. At this point, it is actually irrelevant whether the latter would be ISIDA descriptors or something else, so the ISIDA label was removed. We never include references in the Abstract, as the latter might be published stand-alone, breaking the reference links.

The introduction is extremely long with too many methodological details in the second half. After reading the introduction it is still not clear why GTM and CF were investigated for the docking problem. The goals are not clearly defined: there is a rather long and highly qualified text on "Assessing the convergence" and "a posteriori validation" but it does not tell too much for me. Probably, it is my fault.

Ø  Debatable – I actually checked other recently published papers of ours: with 2 pages of manuscript, this Introduction is… amongst the shortest in that list. We are sorry that this reviewer finds the goals to be not “clearly defined”: there are 3 bullet points for the three goals, and these are (1) reproducibility assessment of the S4MPLE docking simulations, (2) assessment of predictability of native-like poses and (3) ligand prioritization in virtual screening.

Concerning the validation. I cannot appreciate that the 84 ATP site-binders show no significant induced fit modifications on the binding site (line 88, p. 2). Excluding ligands with induced fit obviously results in a biased training/validation.

Ø  Well, this is a proof-of-concept study of CS maps in docking, and, like in all proof-of-concept studies one would like to avoid collateral difficulties. Proper handling of induced fit is still a hot topic – pursuing both goals of CS map methodology development and induced fit handling likely means reaching none of them! Certainly, we would have not be able to support the 10 to 100-fold increase in docking time likely upon enabling site flexibility. Each thing at its time!

The language of the MS is also very confusing at such important points (line 90): "cross‐docking (redocking, for the 1CKP ligand)" was "of" meant instead of "for"?

Ø  True, this was rewritten as follows “A subset of 84 ATP site binders from the PDB, showing no significant induced-fit modifications of the active site was selected. The active site geometry of 1CKP was used throughout this work. The cross-docking (respectively redocking, in the case of the 1CKP ligand) of all the 84 ligands into this 1CKP active site could be achieved with good results.”

At the top of page 3, there is a nice "Methods" section still inside the Introduction.

Ø  Yes, there is. In general, we like to give a brief overview of Methodology in Introduction – the more so here, in a journal relegating Methods after Results… which makes methodology papers really awkward to read.

In the R & D, figs of various landscapes are extensively used. For me, it was impossible to interpret these landscapes and get the accurate (or at least some) physical meaning of the maps or the results in general. I know, the maps do not have such, but still... Again, there are confusing and/or missing definitions, also in the statistics etc.

Ø  OK, this can be… but we would have appreciated knowing what exactly caused confusion, and what is missing (or perhaps hidden in the yet to come Methods section). Authors in general tend to take things unfamiliar to readers for granted, and we did our best to avoid this – without clear feedback we don’t know how to respond here.

"It is thus unclear, whether this GTM model may also be a relevant support for describing CF interaction patterns of ligands other than 1CKP.." (p. 5, line 191) Maybe, this is the only sentence in this MS, I can completely agree.

Ø  There should be some more sentences you might completely agree with: “Cyclin-dependent Kinase 2 (CDK2, e.g. P24941 by Uniprot Accession ID) has been selected for this study” – we swear we did not use any other target, etc. Frankly, this comment was useless and insulting.

The web link on p. 14 does not work.

Ø  It actually does, we just rechecked (there might have been some temporary internet access problem at the time of accession, sorry).